# Risks of Accidents Caused by the Use of Smartphone by Pedestrians Are Task- and Environment-Dependent

**DOI:** 10.3390/ijerph191610320

**Published:** 2022-08-19

**Authors:** Sidney Afonso Sobrinho-Junior, Azriel Cancian Nepomuceno de Almeida, Amanda Aparecida Paniago Ceabras, Carolina Leonel da Silva Carvalho, Tayla Borges Lino, Gustavo Christofoletti

**Affiliations:** 1Applied Biomechanics Laboratory, Institute of Health, Federal University of Mato Grosso do Sul, UFMS, Campo Grande 79060-900, Brazil; 2Faculty of Medicine, Federal University of Mato Grosso do Sul, UFMS, Campo Grande 79060-900, Brazil

**Keywords:** smartphone, cell phone, pedestrian, multitasking behavior, traffic accidents, gait, internet addiction disorder, attention, attentional bias

## Abstract

Using smartphones during a task that requires an upright posture can be detrimental for the overall motor performance. The aim of this study was to determine the risks of accidents caused by the use of smartphones by pedestrians while walking in a controlled (laboratory) and a non-controlled (public street) environment. Two hundred and one participants, 100 men and 101 women, all young adults, were submitted to walking activities while texting messages and talking on the phone. The risk of accident was measured by the time and the number of steps necessary to walk a 20 ft distance. Assessments were performed with no external distractors (laboratory) and on a public street with vehicles, pedestrians, lights, and noises. Multivariate analysis of variance tests provided the main effect of task (using × not using smartphone), environment (laboratory × street), sex (men × women), and interactions. Significance was set at 5%. The results showed that using a smartphone while walking demanded a greater number of steps and time to perform the task (main effect of task: 0.84; *p* = 0.001). The risk of accident was higher on the streets where, due to traffic hazards, pedestrians performed the task faster and with a lower number of steps (the main effect of environment: 0.82; *p* = 0.001). There was no difference of risks between men and women (main effect of sex: 0.01; *p* = 0.225), whether in the laboratory or on the street (main effect of sex × environment: 0.01; *p* = 0.905). The task × environment interaction showed that using a smartphone on the street potentiates risks of accidents of pedestrians (main effect of task × environment: 0.41; *p* = 0.001). In conclusion, using a smartphone while walking can be risky for pedestrians, especially in a traffic environment. People should avoid using their smartphone while crossing streets.

## 1. Introduction

The use of smartphones is rapidly growing around the world. This is happening because several apps are making ordinary tasks easier and faster. For instance, people can order food, go shopping, pay bills, practice exercises, go to the doctor, or event pick up their lab exams with just a few clicks. Today’s mobile phone has many features besides talking or sending messages. All its possibilities are making people more and more attracted to the benefits of the digital world, regardless of the risks they may cause [1,2].

“Digital distraction” defines the use of electronic devices by pedestrians during walking on the street [3]. While smartphones can save time and solve pedestrian’s demands more quickly, they can cause accidents by dividing the attention between the phone and the walking task [4,5].

Previous studies report the risks of using smartphones while walking [6,7,8]. It appears that dual tasking with a smartphone negatively affects gait in young and old adults [9,10]. However, since young adults are the segment of society that has most appropriated smartphones in their lives, pedestrian injury rates for young adults are much higher than those of old adults [11].

Walking is an automatic task where specialized brain circuits coordinate complex patterns of neuromuscular activation [12]. During the use of a smartphone, people’s gaits present shorter strides and step lengths, wider step widths, and lower speeds [13]. In addition, the trunk’s rotation ends up being reduced, affecting the ability to walk and to balance [14,15].

The current literature suggests that gait performance is decreased by an increasing task demand and that specific brain areas are activated according to attentional and energy-optimization challenges [16].

A previous study showed that multitasking with smartphones has different psychological determinants than other forms of multitasking. While people seem to understand the risks of assessing their smartphones on traffic, they cannot avoid using the device while driving cars or crossing the streets [17].

Many of the accidents involving pedestrians are related to subjects’ age and sex. Subjects younger than 35 years old, for example, report more frequent risky behaviors in traffic than other age groups. Women are generally more cautious, risk averse, and compliant road users than men [18]. Generally, a high-risk perception leads to a more cautious behavior [19]. Studying the impact of smartphone use in young adults is important in the face of public health policies.

Until now, most of the studies that have analyzed the impact of smartphone use while walking were performed in laboratories, using a semi-immersive virtual environment with a treadmill. In this scenario, the impact of environment risks such as uneven floors, lights, temperature, weather conditions, and the flow of pedestrians and vehicles, are not considered.

The present study is one of the first to assess the risks of accidents caused by the use of smartphones by pedestrians in a controlled (laboratory) and non-controlled (public street) environment. The target audience is young adults, since this segment has most appropriated smartphone in their lives and because young adults usually report more risky behaviors in traffic than other age groups [11,18]. The authors hypothesized that using a smartphone while walking would increase the risks of accidents of pedestrians, and those risks would be higher when performed with real traffic distractors.

## 2. Materials and Methods

Two hundred and one participants were enrolled in this study. The research was conducted at the Applied Biomechanics Laboratory of the Federal University of Mato Grosso do Sul. The Institutional Review Board approved the research protocol (#4.908.133, CAAE: 47951121.2.0000.0021). All the participants provided written consent prior assessments.

Individuals were included in case they had their own smartphone and were able to stand and walk without assistance. Exclusion criteria comprised cases of neurological or psychiatric disorders, use of orthoses or prostheses in lower limbs, and history of dizziness or recent surgery (<6 months). Additionally, participants were screened for cognitive decline with the mini-mental state examination [20] and with the frontal assessment battery [21]. Normative values were used as eligibility criteria for both instruments [22,23,24].

Sample size was calculated assuming the design of two independent groups (men × women), six measurements (three indoors and three outdoors), with a type 1 error of 5%, power of 80% and a smartphone dual-task effect of 0.26 [25]. The analysis indicated the need of 196 participants. Table 1 details general characteristics of the participants.

Participants underwent three motor tasks: (1) get up from a chair and walk 20 ft without using cell phone; (2) get up from a chair and walk 20 ft while answering a phone call; and (3) get up from a chair and walk 20 ft while texting messages on the cell phone. The order of the tasks was random.

For the talking on the phone task, the participants placed their cell phones in the front pocket of their pants and they were advised to continue to perform the motor task while answering the cell phone call. In this test, one researcher stood beside the subjects (alert in case of falls) and a second researcher stood at a 30 ft distance to call the participants. The chat involved general questions such as food preferences, sport interests, weather conditions, political spotlights, etc.

For the texting message task, the cell phone was positioned inside participants’ front pockets and the subjects were instructed to pick up their cell phone and send the following message while walking: “Good morning, I am going to be late for our appointment.” The sentence was told to the participant at the beginning of the task. 

A 2D digital camera recorded data of the tasks. In this study, risks of accidents were assessed according to time to complete the task and number of steps. If the task was performed with a greater time and number of steps, that means that participants’ attention was centered on the smartphone. If the task was performed with lower time and number of steps, that means that participants’ attention was centered on the walking task.

Evaluations were performed at the Applied Biomechanics Laboratory and on a public street in the city of Campo Grande, state of Mato Grosso do Sul, Brazil. Campo Grande is a large-sized city in the Midwest region of Brazil with approximately 920,000 habitants. The city has ~610,000 vehicles (cars and motorcycles). Private transportation is the most used way of locomotion in the city. Traffic accidents are highly frequent in the city of Campo Grande, especially involving motorcyclists, pedestrians, and motorcycle vs car collisions [26].

The order of the assessments was random for task (walking without using smartphone × walking while talking on the phone × walking while texting messages) and for location (laboratory × street). A software was used for randomization purposes. 

The Applied Biomechanics Laboratory is housed in the Institute of Health, Graduate Program in Movement Sciences of the Federal University of Mato Grosso do Sul. The laboratory is located in an area of approximately 100 ft^2^. The indoor evaluations were controlled in terms of floor regularity, lighting (six 9 W lights), temperature (between 75 and 85 °F), and disturbing sounds (up to 40 dB).

Assessments outdoors involved a 20 ft wide street, with a traffic volume between 250 and 300 vehicles per hour, and no traffic lights. There was a small step (~2 in) between the sidewalk and the street. The chair was positioned on the sidewalk. Participants performed the same tasks as presented in the lab, but now including external distractors (uneven floor, vehicles, pedestrians, weather conditions, noises, etc.). 

The tests were carried out in all weather conditions except rain, snow, and overnight. Researchers realized that these conditions would include other confounding variables (such as holding an umbrella or wearing gloves) that would make smartphone use difficult. Texting messages errors were computed as an indicative of division of attention. 

### Statistical Analysis

The data are reported as mean ± standard deviation. After the normality and homogeneity of the variance were confirmed, the authors ran the repeated measures multivariate analyses of variance (MANOVA) for inferential purposes.

MANOVA were applied in association with Wilk’s Lambda test to verify main effect of task (no smartphone × talking on the phone × texting messages), environment (lab × street), sex (men × women), and interactions.

Univariate analyses of variance (ANOVA) provided detailed assessments for the factors “time to perform the task” and “number of steps”. The crosstab chi-squared test compared texting errors in the lab and on the street. Contrast analyses were used to investigate which task was more challenging to the participants (no cell phone × talking on the phone × texting messages).

Raw data higher than 1.5 interquartile range were identified as outliers, being excluded from the final statistical model. Effect sizes and statistical power are reported [27]. In all analyses, significance was set at 5%.

## 3. Results

Two hundred and one pedestrians (100 men and 101 women, all young adults) were enrolled in this study. Participants were faster and took less steps on the street than in the lab. ANOVA showed that there was no difference between men and women for time (main effect of sex on time: 0.001, power of 6.2%, *p* = 0.750) and number of steps (main effect of sex on number of steps: 0.003, power of 18.0%, *p* = 0.298).

Texting messages showed more challenge than the no phone or talking on the phone tasks (contrast main effect on time: 0.71, power of 99.9%, *p* = 0.001; and contrast main effect on steps: 0.83, power of 99.9%, *p* = 0.001). Table 2 details walking performance of the participants.

The walking pattern of the participants was similar for the time and the number of steps. Figure 1 shows the time (a) and the number of steps (b) according the task (using × not using smartphone, sex (men × women), and environment (lab × public street) factors.

Table 3 presents the MANOVA main effect of sex, task, environment, and interactions. The results showed that using a smartphone while walking increases the risks of accidents of pedestrians (main effect of task: 0.84). Furthermore, pedestrians’ risk of accident was higher on the streets when compared to the laboratory assessment (main effect of environment: 0.82).

There was no difference of risks between men and women (main effect of sex: 0.01), whether in the laboratory or on the street (main effect of sex × environment: 0.01). Real traffic hazards potentiate the risks of accidents of pedestrians (main effect of task × environment: 0.41). There was no significant interaction of the triple combination between the task, environment, and participants’ sex (main effect of task × environment × sex: 0.01).

No participant suffered any fall or dropped their cell phone during the tests. Two researchers stayed alert to avoid traffic incidents.

Fifty four percent of the messages sent in the lab and fifty three percent of the texts sent on the street showed digitation errors. There was no statistical difference between the quantity of errors of the tests performed on the street and at the lab (*p* = 0.122).

## 4. Discussion

Pedestrian distraction is a growing road safety concern worldwide. The aim of this study was to determine the risks of accidents caused by the use of smartphones by pedestrians. The results showed that talking or texting messages while walking increases the risk of accidents, especially on the streets. No differences were seen between men and women in all analyses. We present here the discussion of the findings, which can be of great importance to pedestrians, traffic staff, and public health authorities.

Two hundred and one participants were enrolled in this study. The number of subjects was higher than the minimum stipulated by the sample size calculation. Men and women were similar as for age, cognition, and years of using smartphones. The groups were different for body mass index. In spite of such difference, all the cohort members were categorized with normal weight (values between 18.5 and 24.9 Kg/m^2^). This variable, thus, did not affect subject’s gait patterns [28].

Performing dual tasks is risky for pedestrians. When associated with neurological conditions such as multiple sclerosis, stroke, or Parkinson’s disease, dual tasking becomes more challenging [29,30,31]. This happens because not only the brain connectively is affected, but subjects’ motor functions are also compromised [32].

In the present study, all participants were classified as young adults (aged between 18 and 25 years). Authors focused the analyses on this audience because pedestrian injury rates in young adults are much higher than in older adults [11].

Previous studies report the risks of using a smartphone while walking [33,34]. The differential of this study among the others is that we evaluated participants in a non-controlled environment (public street), seeking to assess the impact of real traffic distractors. 

Men usually present more risky behavior in traffic than women. In spite of women been generally more cautious, risk averse, and compliant road users than males [18], in the present study there was no difference of risk of accidents between men and women.

Pedestrians are subject to minor traumatic injuries, caused, for example, by collisions with other pedestrians, and to more serious situations, where people may not recognize street obstacles, twist their feet, fall down, and even get hit by cars or motorcycles [35,36]. The discussion of the impact of a smartphone on a pedestrian’s health, therefore, is very important.

The main aspect of this study is that it showed that the performance of pedestrians in the lab was different from the performance on the street. While in the lab pedestrians needed more time and steps to accomplish the task, on the street pedestrians performed the tasks faster and with a lower number of steps.

Since most of the studies assess the dual-task effect in a controlled environment (lab), these studies usually identify that pedestrians center their attention on the smartphone, harming the walking task, which ends up with longer time and number of steps.

However, when the task is performed on the street with real traffic hazards, pedestrians centered their attention on the walking task, ignoring the smartphone as a primary focus of attention. That means that the attention of pedestrians depends not only on the type of task (single task without smartphone × dual task with smartphone), but also on the environment (laboratory, a controlled environment × public street, a non-controlled environment). These findings are detailed in Figure 1.

Furthermore, the results showed that texting messages or talking on the phone while walking increases the time and number of steps to accomplish the tasks. This finding corroborates Kim and colleagues [14] and can be interpreted as a result of a more cautious gait pattern while using smartphones. 

Texting messages showed to be more challenging to pedestrians than when talking on the phone or just walking. Authors believe that, although the act of texting messages is increasingly common nowadays, it is unlikely to be as well practiced as walking and talking. Furthermore, for Lamberg and Muratori [37], the increased attentional demands required for texting messages may lead to errors in the otherwise sub-conscious task of walking. This may imply a greater cognitive effort in performing the walking and texting task.

During texting messages, the participant’s visual field is focused on the smartphone, reducing feedback from environmental hazards. Subjects’ upper limbs are holding and texting on the smartphone, which ends up reducing trunk rotation, the ability to walk, and to balance [14,15]. Those aspects may explain the difficulties of participants in performing the texting and walking task.

Authors opted to carry out the tests on a street without traffic lights aiming to stimulate the conflict of action between the smartphone task and possible environmental risks. Participants stayed on the crosswalk and they needed to calculate the risks of crossing the street while using their smartphones.

Hou and colleagues [6] studied pedestrian’s behavior while crossing streets. The authors found that attitudes, intention, and perceived behavioral control are aspects related to a subject’s decision making in traffic. In that way, a street without traffic lights would potentiate the conflicts of action, which was the intention of the authors.

An interesting study showed that pedestrian awareness during walking is lower when associated with cognitively demanding tasks [38]. That means that using smartphones while walking may divide attention between the two tasks. The authors, however, argue that because walking is an automatic task, pedestrians display a general lack of attention while walking and dual tasking may not increase the risks of accidents on the streets.

Our findings showed different conclusions than those of Harms and colleagues [38]. On one hand, we agree that walking and using a smartphone may divide attention between the tasks. On the other hand, pedestrians needed more steps and time to perform the tasks with a smartphone, i.e., subjects increased their support base and decreased their speed to focus the attention on the smartphone task. Those are safety procedures performed unconsciously by pedestrians to avoid accidents.

Performing dual tasks with a smartphone on the streets were more challenging than performing the test in the lab. Due to the risks of traffic accidents, pedestrians finished the tests faster and with a lower number of steps. The task × environment interaction showed that external factors, such as vehicles flow, pedestrians, lighting, uneven floor, street sounds, etc., might play a role in affecting a pedestrian’s safety while using smartphones.

The authors believe that the results of this study should stimulate new public health policies. Traffic authorities, health care professionals, and the general population should be aware of the risks of using cell phones while walking on the streets.

### Limitations

Although the current study provides important information about the risks of using a smartphone while walking, it has some limitations that need to be considered. First, results may be restricted to young adults crossing a 20 ft-wide street without traffic lights.

Second, as the participants were aware of the aim of this study, it is possible that their behavior during the walking tasks affected the results. In other words, pedestrians without been assessed may present different results than the participants of this study.

In that matter, since we compared results in two different scenarios (lab × street) we believe that if a pedestrians’ behavior affected the study, it affected the laboratory and the street assessments on a similar basis, i.e., we believe that the pattern found in this study might be similar to the one found in pedestrians without been evaluated.

Third, the authors used a 2D digital camera to record subjects’ walking pattern. A 3D gait system would provide more details about the impact of smartphone use on spatiotemporal walking parameters. However, since part of this study was performed on the street, a 3D gait system would not be appropriate, as it would capture several “noises” (vehicles, pedestrians, sounds, uneven floor, etc.), affecting the quality of the data.

## 5. Conclusions

This study is one of the first that assessed pedestrian risks while using smartphones on the street. The results provided important findings. First, using a smartphone while walking increases the risks of accidents of pedestrians (affecting both the number of steps and the time to perform the task). Second, there was no difference of risks between men and women, whether in the lab or on the street. Third, real traffic distractors potentiate the risks of accidents of pedestrians, with participants performing the tasks faster and with a lower number of steps. Fourth, texting messages showed to be more challenging to pedestrians than just walking or walking and talking on the phone.

All these findings should stimulate educational campaigns in traffic to inhibit cell phone use during walking.

## Figures and Tables

**Figure 1 ijerph-19-10320-f001:**
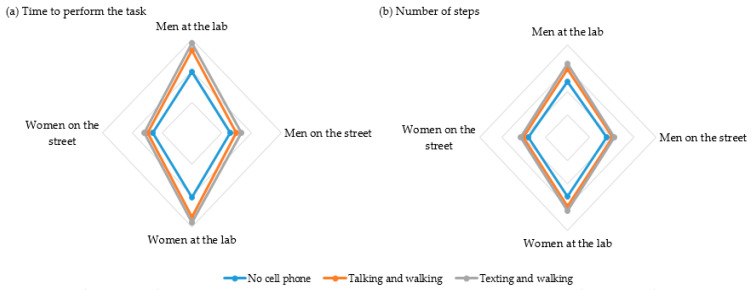
Performance of the participants according to time (**a**), number of steps (**b**), sex, task, and environment.

**Table 1 ijerph-19-10320-t001:** General characteristics of the participants.

Variables	Men	Women	95% Confidence Interval	*p*
Sample size, n	100	101	---	0.994
Age, years	19.9 ± 2.0	20.4 ± 2.1	−0.07 to 1.06	0.090
Body mass index, Kg/m^2^	23.7 ± 4.1	22.3 ± 3.5	−2.8 to −0.6	0.002
Years of using smartphone	9.7 ± 2.7	9.6 ± 2.3	−0.8 to 0.5	0.706
Mini-mental state examination, pts	28.9 ± 1.1	28.8 ± 1.2	−0.4 to 0.2	0.516
Frontal assessment battery, pts	16.6 ± 0.9	16.6 ± 1.0	−0.2 to 0.2	0.932

Data are presented in number of events for categorical variables and mean ± standard deviation for continuous variables. *p* value of the chi-square test for the categorical variables and *p* value of the Student’s *t*-test for the continuous variables.

**Table 2 ijerph-19-10320-t002:** Walking performance of the participants.

Variables	Sex	Task Performed in the Lab			Task Performed on the Street		
		No Cell Phone	Talking on the Phone	Texting Messages	No Cell Phone	Talking on the Phone	Texting Messages
Time, sec	Men	9.8 ± 1.6	13.2 ± 2.6	14.4 ± 2.6	6.5 ± 0.7	7.5 ± 1.3	8.3 ± 1.7
	Women	10.3 ± 1.1	13.4 ± 2.7	14.3 ± 2.7	6.6 ± 0.7	7.5 ± 1.0	8.1 ± 1.4
Steps, n	Men	12.3 ± 1.1	14.7 ± 2.0	16.0 ± 1.9	8.8 ± 0.9	10.1 ± 1.1	10.6 ± 0.9
	Women	12.6 ± 1.3	14.7 ± 1.8	15.8 ± 1.7	8.9 ± 0.9	10.0 ± 1.1	10.6 ± 0.9

Data are presented as mean ± standard deviation.

**Table 3 ijerph-19-10320-t003:** Effect sizes, power, and significance of the multiple analyses of variance tests.

MANOVA Main Effect	Effect Size	Power (%)	*p*
Task (no cell phone × talking × texting)	0.84	99.9	0.001
Environment (lab × street)	0.82	99.9	0.001
Sex (men × women)	0.01	31.9	0.225
Sex × Task interaction	0.02	6.13	0.085
Sex × Environment interaction	0.01	6.5	0.905
Task × Environment interaction	0.41	99.9	0.001
Sex × Task × Environment interaction	0.01	23.8	0.566

## Data Availability

The data presented in this study are available on request from the corresponding author.

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
