# Peer review of "Risks of Accidents Caused by the Use of Smartphone by Pedestrians Are Task- and Environment-Dependent"

_ijerph, 2022, doi:10.3390/ijerph191610320_

Round 1
Reviewer 1 Report
I read your article with much interest.
Paper presents interesting research on the risk of traffic accidents for pedestrians using smartphones. The research seems to have a good experimental structure, well described in the "materials and methods." Although one key question seems unclear to me: How is the risk of having an accident assessed? What is essentially the dependent variable? The time and steps? It seems to me that this part is not adequately described and is central to the whole research.
Also, the introduction presents a very poor scientific literature. In my opinion, the introduction and discussions need to be expanded, otherwise the article gives the impression of being very simple, and arriving at conclusions that are quite trivial and not very " weaved" with the scientific literature. Many authors have dealt with road risk due to cell phone use, such as: Zhang et al. 2020 Fraschetti et al. 2021; Cordellieri et al. 2022; etc.
Thank you very much and good research
Author Response
Dear Reviewer #1. We appreciate your feedback, suggesting important points of improvement of the manuscript. We explain here each point suggested and our response, detailing the inclusions that we have made in the text:
Suggestion 1: I read your article with much interest. Paper presents interesting research on the risk of traffic accidents for pedestrians using smartphones. The research seems to have a good experimental structure, well described in the "materials and methods”
Authors’ response: Thank you for your feedback. Your external review highlights the quality and the originality of the study, and shows important points needing improvement. We choose the IJERPH because we think that the journal will give a good visibility of the manuscript in the scientific community.
Suggestion 2: One key question seems unclear to me: How is the risk of having an accident assessed? What is essentially the dependent variable? The time and steps? It seems to me that this part is not adequately described and is central to the whole research.
Authors’ response: We thank the reviewer for highlighting this point. In this study risk of accident was assessed in time and number of steps. That information was not clear neither on the abstract section nor in the text. We included more information in this new version of the manuscript, as you can see:
Abstract section: “Risk of accident was measured by the time and number of steps necessary to walk a 20ft distance. Assessments were performed with no external distractors (laboratory) and on a public street, with vehicles, pedestrians, lights and noises”….. “Results showed that using smartphone while walking demands greater number of steps and time to perform the task (main effect of task: 0.84; p=0.001). The risk of accident was higher on the streets, where, due to traffic hazard, pedestrians performed the task faster and with lower number of steps (main effect of environment: 0.82; p=0.001).”
Methods section: “Risks of accidents were assessed according to time to complete the task and number of steps. If the task was performed with a greater time and number of steps, that means that participants’ attention was centered on the smartphone. If the task was performed with lower time and number of steps, that means that participants’ attention was centered on traffic hazard.”
Suggestion 3: Also, the introduction presents a very poor scientific literature. In my opinion, the introduction and discussions need to be expanded, otherwise the article gives the impression of being very simple, and arriving at conclusions that are quite trivial and not very " weaved" with the scientific literature. Many authors have dealt with road risk due to cell phone use, such as: Zhang et al. 2020 Fraschetti et al. 2021; Cordellieri et al. 2022; etc.
Authors’ response: We agree with the reviewer that both introduction and discussion sections needed improvements. We included all references suggested by Reviewer #1 and #2, and more text (highlighted in this new version of manuscript with “Word processing changes”). We believe that the manuscript is much more complete ain this new version. We thank Reviewer 1 for his/her time in reviewing this study.
Reviewer 2 Report
The study investigates the risks of accidents caused by using smartphones by pedestrians while walking. It is a good paper but should address the following issues before publication.
The title is general. At least, I suggest including the road user that was investigated in the title (pedestrian).
Following my previous comment, I can see that the people were young. You should also indicate it in the title or at least in the abstract.
In the abstract, briefly explain how the risks of accidents were measured. In other words, the methodology should be outlined in the abstract.
Various studies such as 10.1016/j.aap.2021.106050 have shown the interaction between pedestrian behaviors and crashes. I suggest you read them more carefully.
A detailed explanation/justification is required to support the current study's findings. What is the logic behind these findings? What is the application of these findings?
Following my previous comment, some findings question the value of the other conclusions. For example, when you say the risk of accident was higher on the streets compared to the laboratory assessment: if we consider this finding in light of the crowded street, what valuable point has it determined? Further, what is the value of laboratory results when they cannot simulate the natural conditions?
When people were aware of the study, how can you claim that their behavior was not affected?
The analyses used in this study are elementary and limited. This journal is among the high-quality journals. So, I expected to see more sophisticated analyses/models.
Author Response
Dear Reviewer #2. We thank your feedback, suggesting important points of improvement of the manuscript. We explain here each point suggested and our response, detailing the inclusions that we have made in the text:
Suggestion 1: The study investigates the risks of accidents caused by using smartphones by pedestrians while walking. It is a good paper but should address the following issues before publication.
Authors’ response: We appreciate all suggestions. We agreed with all points suggested by Reviewer #1 and #2, as you can see in this response letter.
Suggestion 2: The title is general. At least, I suggest including the road user that was investigated in the title (pedestrian).
Authors’ response: We agree with reviewer’s suggestion and we included “pedestrians” in the title. With this inclusion, the title became “Risks of Accidents Caused by the Use of Smartphone by Pedestrians are Task and Environment Dependent”.
Suggestion 3: Following my previous comment, I can see that the people were young. You should also indicate it in the title or at least in the abstract.
Authors’ response: We agree with the reviewer and we included the following text in the abstract section: “Two hundred and one participants, 100 men and 101 women, all young adult, were submitted to….”.
Suggestion 4: In the abstract, briefly explain how the risks of accidents were measured. In other words, the methodology should be outlined in the abstract.
Authors’ response: We agree with the reviewer that this point was not clear in the original version of the manuscript. We included the following text in the abstract section to address reviewers’ suggestion: “…Risk of accident was measured by the time and number of steps necessary to walk a 20ft distance. Assessments were performed with no external distractors (laboratory) and on a public street, with vehicles, pedestrians, lights and noises.” In addition, we included: “…Results showed that using smartphone while walking demand greater number of steps and time to perform the task (main effect of task: 0.84; p=0.001). The risk of accident was higher on the streets, where, due to traffic hazard, pedestrians performed the task faster and with lower number of steps (main effect of environment: 0.82; p=0.001)…”
Suggestion 5: Various studies such as 10.1016/j.aap.2021.106050 have shown the interaction between pedestrian behaviors and crashes. I suggest you read them more carefully.
Authors’ response: Both reviewers suggested the inclusion of references aiming to strength the Introduction and Discussion sections. We included several articles and texts, highlighted in this new version of the manuscript with the “tracking changes of the Microsoft Word”. The suggested study (10.1016/j.aap.2021.106050) was very interesting and it was included in our review.
Suggestion 6: A detailed explanation/justification is required to support the current study's findings. What is the logic behind these findings? What is the application of these findings?
Authors’ response: We agree that the discussion section needed to be improved. We included the following text to explain the finding: “The main aspect of this study showed that the performance of pedestrians in the laboratory is different from the performance on the street. Whereas in the laboratory pedestrians needed more time and number of steps to accomplish the task, on the street the pedestrians performed the tasks faster and with lower number of steps.
Since most of the studies assess dual-task effect in a controlled environment (laboratory), these studies usually identify that pedestrians centered their attention on the smartphone, harming the walking task, which ends up with longer time and number of steps.
However, when the task is performed on the street, with real traffic hazards, pedestrians centered their attention on the walking task, ignoring the smartphone as primary focus of attention, and performing the walking task faster.
Suggestion 7: Following my previous comment, some findings question the value of the other conclusions. For example, when you say the risk of accident was higher on the streets compared to the laboratory assessment: if we consider this finding in light of the crowded street, what valuable point has it determined? Further, what is the value of laboratory results when they cannot simulate the natural conditions?
Authors’ response: Dear reviewer. We believe we have answered this question earlier (suggestion 6). The importance of the laboratory was to show that in a relative controlled environment, pedestrians centered their attention on the smartphone. In a more risky situation (public street with real traffic hazards), pedestrians centered their attention on the walking task, changing the focus of their primary attention.
Suggestion 8: When people were aware of the study, how can you claim that their behavior was not affected?
Authors’ response: We agree with the reviewer that the behavior of pedestrians may have affected the results. However, since we compared results in two different scenarios (laboratory × street) we believe that, if this factor affected the study, it affected on similar basis. We included that information in the limitation section.
Suggestion 9: The analyses used in this study are elementary and limited. This journal is among the high-quality journals. So, I expected to see more sophisticated analyses/models.
Authors’ response: We agree with the reviewer about IJERPH, which is among the high-quality journals. We choose the IJERPH because we think that the journal will give a good visibility of the manuscript in the scientific community. However, we would like to respectfully disagree with reviewer about the saying that the statistical analyses are elementary and limited. We are finishing a systematic review on this thematic, and we realized that most of the studies performed similar analyses as presented in this study. In a manner of fact, they do not include neither effect size nor power – which are detailed in this study highlighting group, task and interaction effects. MANOVA is an important statistics that allows complex findings. Thus, we respectfully disagree with the reviewer and we think that our statistical analysis bring much more results than similar published articles. However, due to the suggestion of reviewer #2, we included a figure comparing the results in terms of time to perform the task, number of steps,sex, task and environment. Please see figure 1 in the text.
Round 2
Reviewer 2 Report
No further comment
This manuscript is a resubmission of an earlier submission. The following is a list of the peer review reports and author responses from that submission.